# A “Pro-Asp-Thr” Amino Acid Repeat from *Vibrio* sp. QY108 Alginate Lyase Exhibits Alginate-Binding Capacity and Enhanced Soluble Expression and Thermostability

**DOI:** 10.3390/ijms25115801

**Published:** 2024-05-26

**Authors:** Zheng Fu, Fengchao Zhang, Hainan Wang, Luyao Tang, Wengong Yu, Feng Han

**Affiliations:** 1Key Laboratory of Marine Drugs (Ministry of Education), Shandong Provincial Key Laboratory of Glycoscience and Glycoengineering, School of Medicine and Pharmacy, Ocean University of China, Qingdao 266003, China; 2Laboratory for Marine Drugs and Bioproducts of Qingdao National Laboratory for Marine Science and Technology, Qingdao 266237, China

**Keywords:** alginate lyase, alginate-binding module, multi-domain, soluble expression

## Abstract

Alginate lyases cleave the 1,4-glycosidic bond of alginate by eliminating sugar molecules from its bond. While earlier reported alginate lyases were primarily single catalytic domains, research on multi-module alginate lyases has been lfiguimited. This study identified VsAly7A, a multi-module alginate lyase present in *Vibrio* sp. QY108, comprising a “Pro-Asp-Thr(PDT)” fragment and two PL-7 catalytic domains (CD I and CD II). The “PDT” fragment enhances the soluble expression level and increases the thermostability and binding affinity to the substrate. Moreover, CD I exhibited greater catalytic efficiency than CD II. The incorporation of PDT-CD I resulted in an increase in the optimal temperature of VsAly7A, whereas CD II displayed a preference for polyG degradation. The multi-domain structure of VsAly7A provides a new idea for the rational design of alginate lyase, whilst the “PDT” fragment may serve as a fusion tag in the soluble expression of recombinant proteins.

## 1. Introduction

Alginate, an acidic linear hetero-polyuronic acid, consists of β-D-mannuronate (M) and its C5 epimer α-L-guluronate (G) linked via β-1,4-glycoside bonds and arranged in three different types, namely poly-β-D-mannuronate (polyM), poly-α-L-guluronate (polyG), and alternating or random heteropolymer (polyMG) [1]. It is widely utilized for its thickening, gelling, and biosecurity properties in the biomedical science and engineering fields [2].

Alginate lyases (EC 4.2.2.-) catalyze the cleavage of 1,4-glycoside bonds by elimination reactions and form unsaturated C4–C5 double bonds at non-reducing ends [3], as illustrated in Figure 1A. Based on differences in their primary structure, alginate lyases can be categorized into polysaccharide lyase families 5, 6, 7, 8, 14, 15, 17, 18, 31, 32, 34, 36, 39, and 41 [4,5]. In addition, carbohydrate-active enzymes (CAZymes), such as PLs (polysaccharide lyases), GHs (glycoside hydrolases), and CEs (carbohydrate esterases), generally possess the components of non-catalytic carbohydrate-binding modules (CBMs) that fold independently into discreet functional units [6,7,8,9,10,11,12,13,14]. Previous studies have stated that CBMs can recognize polysaccharides by substrate-specific affinity [15] and enhance catalytic activity by substrate targeting and proximity effects [16,17]. Moreover, they can potentiate the secretion [18] and thermostability [10] of the recombinant fusion protein.

In this study, an atypical multi-module alginate lyase, VsAly7A, was identified from the marine bacterium *Vibrio* sp. QY108. VsAly7A contained an N-terminal “Pro-Asp-Thr(PDT)” fragment followed by two PL-7 catalytic domains (CD I and CD II). The full-length (FL) of VsAly7A and a range of truncated mutants (PDT-CD I, PDT-CD II, CD I-CD II, PDT, CD I, CD II, and GST-PDT) were expressed and characterized to determine the mechanisms by which each domain functioned. The “PDT” fragment played a central role in binding alginate and increasing the soluble expression of recombinant proteins in *E. coli.* Moreover, CD I was identified as a major catalytic domain in VsAly7A, whereas CD II had a distinct substrate specificity. This research on the function of the different domains in VsAly7A offered valuable insights into their relationship with three modules in carbohydrate-active enzymes.

## 2. Results and Discussion

### 2.1. Alginate-Lyase-Encoding Gene Sequence Analysis

Gene *vsaly7A* encodes a protein, VsAly7A, with 603 amino acid residues. It has a theoretical isoelectric point (pI) of 4.45 and a theoretical molecular weight of 65,268.07 Da. According to the sequence analysis (BLASTp/CD-search/CDART) in NCBI and the prediction of the signal peptide using SignalP 5.0, VsAly7A was predicted to contain a signal peptide (Met1-Cys19) at its N-terminal and double alginate-lyase catalytic domains (CD I and CD II) at the C-terminal, as illustrated in Figure 1B.

Alginate lyases containing two catalytic domains have been reported in earlier studies, including Aly7B from *Vibrio* sp. W13 [19], AlyA OU02 from *V. splendidus* OU02 [20], AlyC8 from *Vibrio* sp. C42 [21] and AlyC6’ from *Vibrio* sp. NC2 [22]. Both catalytic domains of VsAly7A were classified within the PL-7 family and were separated by a linker (8 amino acid residues), but the identity between these two catalytic domains was merely 23.32%. Thus, a neighbor-joining tree was constructed using VsAly7A-CD I and -CD II amino acid sequences and PL-7 alginate lyase amino acid sequences to determine its phylogenetic position. The results of the homologous sequence alignment and phylogenetic analysis (Figure 2) indicated that VsAly7A-CD I is a member of the PL-7 subfamily 6; however, VsAly7A-CD II could not be categorized into any subfamily of PL-7. In addition, as displayed in Figure 3A and 3B, both domains contained three highly conserved PL-7 regions, termed RS/T/NEL/VR, QI/VH, and YFKAGV/IY, which contribute to the substrate recognition and catalytic reactions [1,23,24].

Additionally, a unique amino acid sequence was detected between the signal peptide and the catalytic domain I (CD I). It had a length of 54 amino acid residues and nine “Pro-Asp-Thr” repeat fragments (Pro34-Tyr61). We termed this sequence as “PDT” fragment.

### 2.2. The Function of Domains in VsAly7A

To determine the function of the domains in VsAly7A, the full-length of VsAly7A and six truncated mutants was sequentially designed, cloned, expressed, and purified (Figure 4A). Overexpression of all encoding genes was achieved in the pET24a (+)/*E. coli* BL21(DE3) system. Using Ni-affinity chromatography, the recombinant proteins were purified to homogeneity and analyzed using SDS-PAGE (Figure 4B).

Upon optimizing the fermentation conditions for each recombinant protein, the soluble expression levels were found to vary among all the recombinant proteins (Figure 4C and Table 1). Grouped pairwise comparison and SDS-PAGE analysis were conducted to assess the yield of soluble recombinant proteins (FL vs. CD I-CD II; PDT-CD I vs. CD I; PDT-CD II vs. CD II), and the results demonstrated that recombinant proteins lacking the “PDT” fragment were more likely to form inclusion bodies. Likewise, another “PDT” fragment fusion protein was constructed and expressed, which exerted a positive effect on promoting soluble expression. Thus, the “PDT” fragment played a vital role in promoting soluble protein expression and acted as a tag element for recombinant protein to improve soluble protein expression.

Additionally, the specific activities of various enzyme constructs were assessed, with the full-length enzyme displaying an activity of 302.65 U/mg, that of PDT-CD I exerted an activity of 1955 U/mg, and that of PDT-CD II exhibiting an enzyme activity of 70.13 U/mg. Interestingly, CD I-CD II (31.42 U/mg) exhibited a specific activity toward sodium alginate. The aforementioned observations inferred that the inclusion of the “PDT” fragment promoted the enzymatic activity of the recombinant protein. However, the truncated PDT-CD I showed 6.5-fold higher activity than the full-length enzyme, suggesting that CD I was the domain that played a major catalytic activity, whereas CD II seemed to have an inhibitory effect on full-length enzyme activity.

### 2.3. Biochemical Characterization of VsAly7A and Its Truncated Mutants

The optimal temperature (T_opt_) of VsAly7A and PDT-CD I was 40 °C, while that of CD I-CD II was 30 °C (Figure 5A). In addition, after incubating the recombinant proteins at 30 °C for 1 h, PDT-CD I retained 87% of its activity, VsAly7A retained 55%, PDT-CD II retained 42%, and CD I-CD II retained 19% (Figure 5C). Moreover, PDT-CD I exhibited a remarkable 36% retention of its maximal activity following storage at 50 °C for 1 h, in contrast to other recombinant proteins, which experienced a loss of activity. PDT-CD I and PDT-CD II have different T_opt_ and temperature stability (T_m_) values (PDT-CD I vs. PDT-CD II), which were dependent on the properties of different catalytic domains. Further, this was consistent with the T_opt_ of VsAly7A and PDT-CD I, but the T_m_ of VsAly7A was 10 °C less than PDT-CD I. This showed that the in case of CD1, the unfolding of global structure is concurrent with the loss of an active site as its T_opt_ and T_m_ were almost same. In contrast, the T_opt_ of VsAly7A was 10 °C more than its T_m_, suggesting that its active site was collapsing before the unfolding of overall structure. In addition, The T_opt_ and T_m_ of VsAly7A were better than those of CD I-CD II. These findings provide compelling evidence that PDT-CD I possesses superior thermal stability to both the full-length enzyme VsAly7A and other truncated mutants.

At the same time, the optimal pH for recombinant VsAly7A and the truncated mutants was found to be 8.6 (Figure 5B). They were stable in a pH range between 8.0 and 10.6 in different buffers (Figure 5D). According to prior investigations, alginate lyases exhibit optimal pH and stability in a marginally alkaline environment [25].

Figure 5E depicts the impact of NaCl on enzymatic activities. The results indicated that all enzymes displayed elevated activity within the 0.1–1 M NaCl range, while a markable decrease in activity was observed in the absence of NaCl, suggesting that NaCl is a potent activator of VsAly7A and its truncated mutants. As anticipated, the salt-activating nature of these enzymes protects against harsh, high-salt marine environments.

The enzyme kinetics of VsAly7A and the truncated mutants towards alginate are summarized in Table 1. The *K_m_* value of VsAly7A (0.62 mM) was similar to that of PDT-CD I (0.57 mM) and PDT-CD II (0.16 mM), whereas that of CD I-CD II was considerably higher (4.64 mM), signaling that the full-length enzyme, truncated PDT-CD I, and PDT-CD II, exhibited a similar affinity to sodium alginate, but that of CD I-CD II was significantly lower than other recombinant proteins. Importantly, the *kcat*/*K_m_* value of PDT-CD I was higher than that of the full-length enzyme VsAly7A (3.3-fold), PDT-CD II (8.1-fold), and CD I-CD II (116.7-fold). These results were consistent with the above-mentioned findings.

### 2.4. Substrate Specificity and Reaction Products of VsAly7A and Its Truncated Mutants

In order to ascertain substrate specificity, enzymatic activity was evaluated using 0.30% (*w*/*v*) sodium alginate, polyM, and polyG. The results displayed in Figure 6A reveal that VsAly7A exhibited a preference for polyM, with relative activity values of 100.00 ± 1.06%, 97.21 ± 0.60%, and 51.94 ± 0.74% for sodium alginate, polyM, and polyG, respectively. Notably, the relative activity of PDT-CD I and CD I-CD II towards the aforementioned substrates was comparable to that of full-length VsAly7A. Nevertheless, the substrate specificity of PDT-CD II suggested that it was a polyG-preferred alginate lyase. This result was in line with the substrate specificity of domains in Aly7B [19], AlyA OU02 [20], AlyC8 [21] and AlyC6’ [22], suggesting that the two catalytic domains play distinct roles in degrading diverse substrates.

Afterward, thin-layer chromatography (TLC) analysis was employed in order to examine the degradation mechanism of VsAly7A and its soluble truncated mutants. The results shown in Figure 6B indicate that during the initial phase of the degradation process, both VsAly7A and the truncated mutants generated a minimal amount of low-molecular-mass oligosaccharides. Over time, the amount of alginate oligosaccharides progressively increased. These results conjointly indicated that VsAly7A displayed an endo-type degradation mode that was unaltered by truncation.

To analyze the end products of VsAly7A and its truncated mutants, ESI-MS and FPLC were carried out using a superdex peptide 10/300 column. The results, as portrayed in Figure 6C, revealed that the end products of VsAly7A comprised di-, tri-, tetra-, penta-, and hexa-saccharides, whereas those of PDT-CD I were unchanged. However, attributed to the low enzyme activity and poor thermal stability of PDT-CD II and CD I-CD II, it was challenging to determine their respective end products.

### 2.5. “PDT” Fragment: A Novel Alginate-Binding Module

To verify that the “PDT” fragment was a novel alginate-binding module, an attempt was made to express the “PDT” fragment in isolation, but our attempt failed after induction. Therefore, the recombinant protein PDT-GST, which is a “PDT” fragment combined with a GST tag, was designed (Figure 7A). Following this, the co-incubation of recombinant protein with alginate gel beads was used to analyze the affinity of the “PDT” fragment to sodium alginate. Comparing the free protein concentration of PDT-GST and GST, as shown in Figure 7B, suggested that GST-PDT could combine with alginate, thereby insinuating that the “PDT” fragment was a novel alginate-binding module.

## 3. Materials and Methods

### 3.1. Materials and Strains

The standard sodium alginate (Food grade, M/G ratio: 3/5) was procured from Gather Great Ocean Algae Industry Group Co., Ltd. (Qingdao, China), while poly-M and poly-G (concentration ~90%) were acquired from Qingdao BZ Oligo Biotech Co., Ltd. (Qingdao, China). *E. coli* strains were cultured in Luria–Bertani (LB) medium supplemented with 30 μg/mL kanamycin, and all recombinant proteins were expressed in BL21 (DE3) (TaKaRa, Dalian, China) cells.

### 3.2. Sequence Analysis of Alginate-Lyase-Encoding Gene

SignalP 5.0 Server (http://www.cbs.dtu.dk/services/SignalP/ (accessed on 8 January 2020)) was used to predict the presence of an N-terminal signal peptide in VsAly7A. ExPASy ProtParam (https://web.expasy.org/compute_pi/ (accessed on 8 January 2020)) was utilized to calculate the theoretical isoelectric point (pI) and molecular weight (Mw) of VsAly7A and its truncated mutants. Conserved Domains Search (https://www.ncbi.nlm.nih.gov/Structure/cdd/wrpsb.cgi/ (accessed on 8 January 2020)) and Conserved Domain Architecture Retrieval Tool (CDART, https://www.ncbi.nlm.nih.gov/Structure/lexington/lexington.cgi/ (accessed on 8 January 2020)) were employed to identify the conserved domain in VsAly7A. Clustal Omega (https://www.ebi.ac.uk/Tools/msa/clustalo/ (accessed on 8 January 2020)) and ESPript 3.0 (http://espript.ibcp.fr/ESPript/ESPript/ (accessed on 8 January 2020)) were used for multiple protein sequence alignment. The molecular Evolutionary Genetics Analysis (MEGA) program, version 7.0 [26] (https://www.megasoftware.net/ (accessed on 8 January 2020)), was used to construct the phylogenetic tree via the bootstrapping maximum likelihood method. The model of VsAly7A was predicted using AlphaFold2 (https://cosmic-cryoem.org/tools/alphafold2/ (accessed on 18 July 2022)) and visualized by PyMOL. VsAly7A was uploaded to GenBank under Accession Number OR921196.

### 3.3. Cloning, Expression, and Purification of Recombinant VsAly7A and the Truncated Mutants

Based on the annotated genome sequence of *Vibrio* sp. QY108, gene-specific primer pairs with *Nde I*/*Xho I* were designed, as presented in Table 2. Subsequently, the gene fragments were integrated into plasmid pET-24a (+) to construct the pET expression vector of VsAly7A full-length (FL) and a range of truncated mutants. PDT-CD II is not a naturally truncated mutant, so four primers were designed, and gene splicing was performed using fusion PCR technology for the experiment. pET-24a-GST-PDT and pET-24a-GST were synthesized by GENEWIZ, Inc. (Suzhou, China).

The competent *E. coli* BL21 (DE3) was transformed with the recombinant plasmids and subsequently cultured in LB medium supplemented with 30 μg/mL kanamycin at 37 °C. Protein expression was induced at OD600 ≈ 0.5 with 0.1 mM isopropyl-β-thiogalactoside (IPTG) and incubated at 18 °C for 24 h. The harvested cells were then disrupted using a high-pressure cell disrupter (JNBIO, Guangzhou, China), and the resulting supernatant was collected after centrifugation to discard cell membranes and insoluble materials. The soluble fractions were purified via the HisTrap^TM^ HP column (Cytiva, Uppsala, Sweden). Thereafter, the purity and molecular weight of the resulting proteins were assessed through 12.5% SDS-PAGE, while proteins were quantified using the bicinchoninic acid assay kit (Vazyme Biotech Co., Ltd., Nanjing, China).

### 3.4. Enzymatic Activity Assay of Recombinant VsAly7A and Truncated Mutants

The experimental procedure involved the introduction of appropriately diluted VsAly7A or truncated mutants (100 μL) to a reaction system (1 mL) containing 0.3% (*w*/*v*) standard sodium alginate substrate (900 μL) in 0.02 M phosphate buffer with 0.3 M NaCl at a pH of 7.3. Next, the mixture was incubated at 40 °C for 10 min, following which the enzymatic activities were estimated by quantifying the escalation in A235 using a UH5300 spectrophotometer (Hitachi High-Technologies Co., Tokyo, Japan), consequent to the emergence of Δ-4,5-unsaturated double bonds during the process of degradation. A single unit of lyase activity (U) was defined as the quantity of enzyme that generated a rise of 0.1 A235 units per minute.

The present study examined the kinetic parameters of VsAly7A and its truncated mutants using various concentrations of standard sodium alginate (ranging from 0.01% to 0.8% (*w*/*v*)). The Michaelis–Menten equation was employed to calculate these parameters, and nonlinear repression plots were generated using GraphPad Prism version 8.0.0 (San Diego, CA, USA).

### 3.5. Biochemical Characterization of Recombinant VsAly7A and Truncated Mutants

The impact of pH on the enzymatic activities of VsAly7A and its truncated mutants was investigated using various buffers, namely 0.02 M Na_2_HPO_4_–citric acid (pH 3.0–8.0), 0.02 M Tris-HCl (pH 7.05–8.95), 0.02 M Na_2_HPO_4_–NaH_2_PO_4_ (pH 6.0–8.0), and 0.02 M glycine-NaOH (pH 8.6–10.6). Furthermore, the pH stability of both VsAly7A and its truncated mutants was assessed in the aforementioned solvents following incubation at 4 °C for 12 h. Furthermore, the degradation system was conducted within a temperature range of 0 °C to 60 °C to determine the optimal catalytic temperature of both VsAly7A and its truncated mutants. The thermostability of these enzymes was assessed by preincubation with a phosphate buffer (0.02 M, pH 7.3) at various temperatures (0–70 °C) for one hour, followed by an enzyme assay at 40 °C. Enzymatic activities of VsAly7A and its truncated mutants were assessed in the presence of 1 mM cation ions or chelators to investigate the impact of different metal ions and chelators. Additionally, substrate specificities were determined using polyM and polyG, sourced from Bozhihui Biological Technology Co., Ltd. in Qingdao, China. The relative activity was calculated by comparing it to the maximal enzymatic activity (considered 100%) in these experiments. All assays were performed in triplicates, and enzymatic activity was expressed as mean ± standard deviation (SD).

### 3.6. Analysis of Reaction Pattern of Recombinant VsAly7A and its Truncated Mutants

Thin-layer chromatography (TLC) was employed to assess the degradation pattern of VsAly7A and its truncated mutants. Briefly, a mixture of 10 mL 0.3% (*w*/*v*) standard sodium alginate and 1 mL of VsAly7A or its truncated mutants (10 U) was incubated at the optimal catalytic temperature for varying durations of 0, 5, 10, and 30, or 60, 180, and 300 min. The degradation fractions of VsAly7A and its truncated mutants were separated using an aluminum TLC plate (TLC Silica gel 60 F254, Merck, Beijing, China) and a TLC running buffer (1-butanol/formic acid/water, 4:6:1, by vol.). The plates were coated with the diphenylamine-aniline-phosphate (DPA) reagent and heated to 130 °C until a distinct color was observed.

### 3.7. Analysis of End Products of Recombinant VsAly7A and PDT-CD I

The AKTA FPLC chromatography system (GE Healthcare Life Sciences, Pittsburgh, PA, USA) and negative-ion ESI-MS were utilized to analyze the end product of VsAly7A and PDT-CD I. Specifically, 1 mL of VsAly7A or PDT-CD I (10 U) was incubated with 9 mL of 0.3% (*w*/*v*) standard sodium alginate at 30 °C overnight. The sample was fractionated through size-exclusion chromatography (SEC) using a Superdex peptide 10/300 GL column (GE Health, Chicago, IL, USA) and monitored for changes in A235 using an AKTA purifier. The mobile phase consisted of 0.2 M (NH_4_)_2_CO_3_ buffer. Following desalination, the molecular masses of the samples were analyzed using negative-ion electrospray ionization mass spectrometry (ESI-MS).

### 3.8. Alginate Gel Bead Preparation and Analysis of Binding Capabilities of Recombinant GST and GST-PDT

The preparation of the alginate gel bead was carried out in accordance with the previously outlined methodology (25). Specifically, 1% (*w*/*v*) sodium alginate was dissolved in 20 mM phosphate buffer containing 300 mM NaCl (pH 7.3). Then, the sodium alginate solution was poured into a 10 mL syringe attached with a 30-gauge needle and added dropwise into a gently agitated 0.2 M calcium chloride solution. The formed gel beads were maintained in the solution for 2 h. The final resulting beads (size about 150~300 μm) were discarded and thereupon rinsed several times with deionized water.

Subsequently, the gel beads were submerged in a 20 mM phosphate buffer containing 300 mM NaCl (pH 7.3), and 10 nmol of either GST or GST-PDT were introduced to each tube containing 10 alginate gel beads. The amalgamation was then refrigerated at 4 °C for 3 h, and the supernatant was collected through centrifugation (12,000 rpm, 10 min, 4 °C). Alterations in protein concentration pre- and post-treatment were calculated using the BCA protein assay kit (Vazyme Biotech Co., Ltd., Nanjing, China).

## 4. Conclusions

In this study, an alginate lyase, VsAly7A, which contained the first reported “PDT” fragment and two catalytic domains, was discovered. Surprisingly, the “PDT” fragment was essential for maintaining the soluble expression and high enzymatic activity of recombinant VsAly7A and its truncated mutants but had no significant effect on other enzymatic properties such as the pH, pH stability, NaCl concentration, substrate specificity, degradation mode, end product and so on. CD I was identified as the main catalytic domain of VsAly7A, while the substrate specificity of CD II was distinct from that of CD I. Further experiments revealed that the “PDT” fragment selectively bound to alginate gel beads. Our upcoming work will be focused on investigating the role of the “PDT” fragment on the soluble expression of other recombinant proteins and the enhancement of alginate lyase activity.

## Figures and Tables

**Figure 1 ijms-25-05801-f001:**
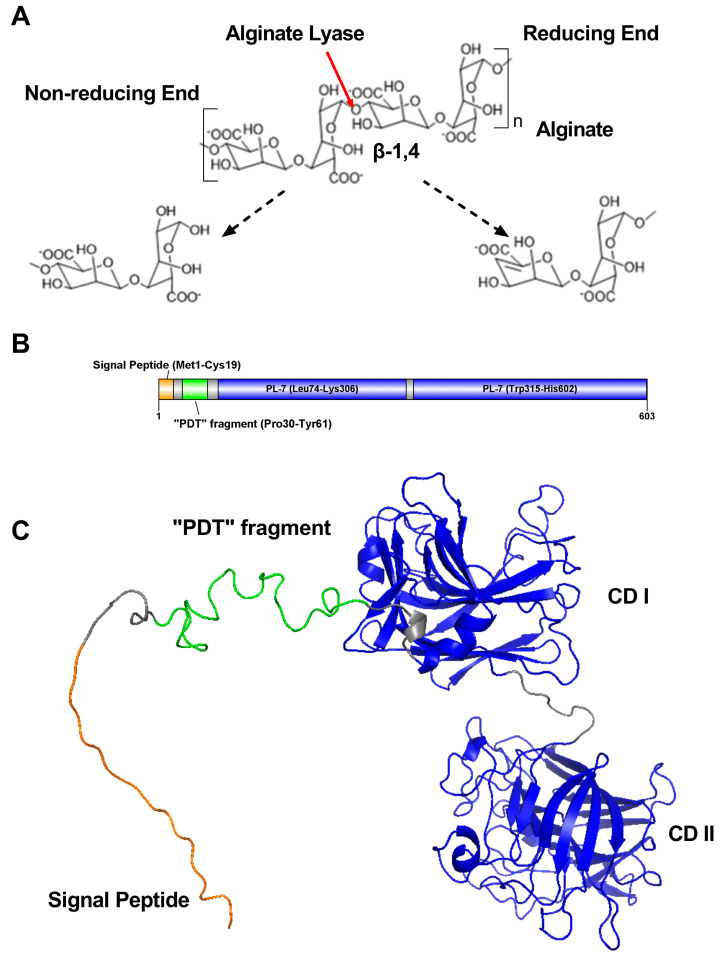
Sequence analysis of the alginate lyase VsAly7A. (**A**) Mechanism of alginate lyase degradation. (**B**) Domain structure of FL VsAly7A. (**C**) Phylogenetic analysis of VsAly7A-CD I, -CD II, and other PL7 lyases from different subfamilies.

**Figure 2 ijms-25-05801-f002:**
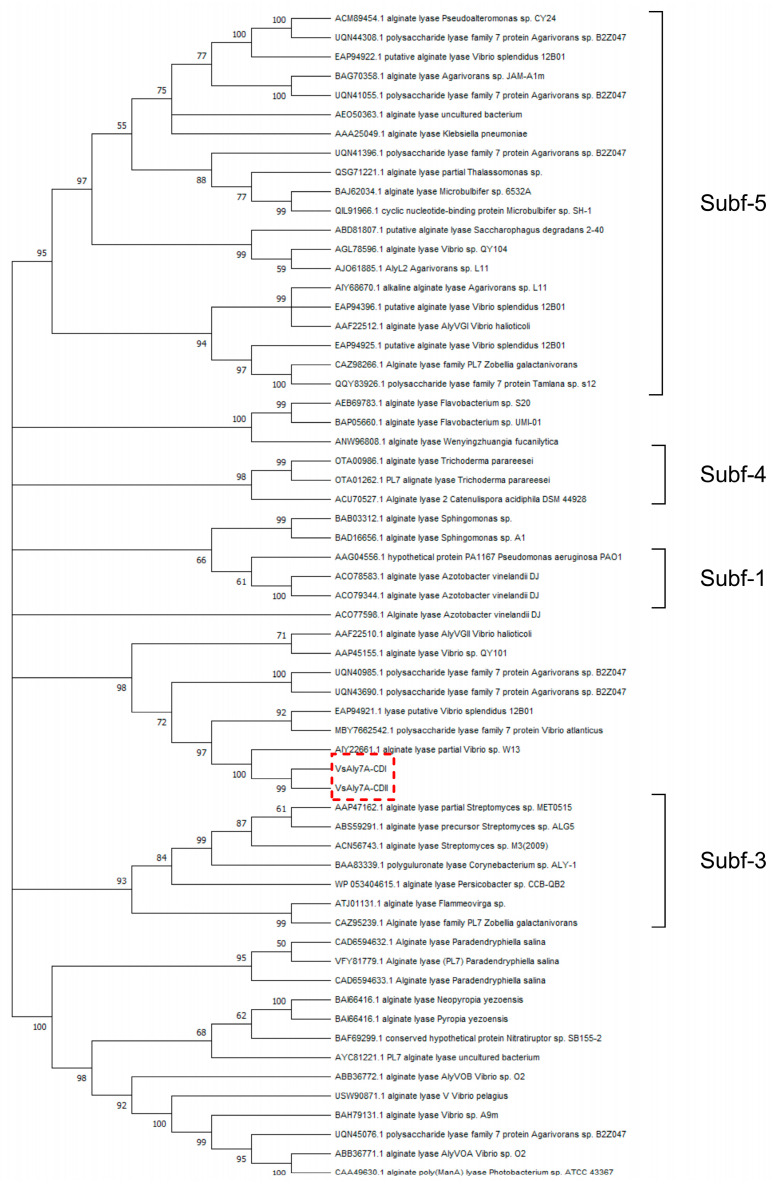
Phylogenetic analysis of VsAly7A-CD I, -CD II, and other PL7 lyases from different subfamilies.

**Figure 3 ijms-25-05801-f003:**
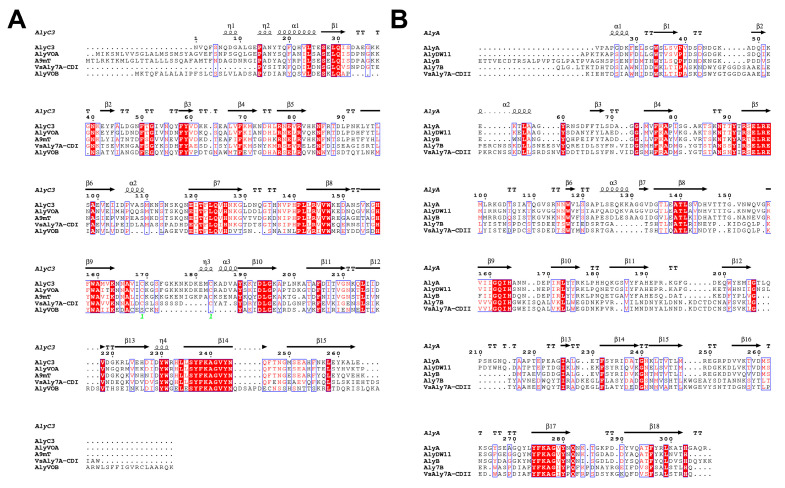
Comparison of the partial amino acid sequences of VsAly7A-CD I (**A**) and -CD II (**B**) from PL7 alginate lyases. AlyC3 (PDB code: 7C8F) from *Psychromonas* sp. C-3, AlyVOA (ABB36771), AlyVOB (ABB36772) from *Vibrio* sp. O2, A9mT (BAH79131) from *Vibrio* sp. A9m, AlyA (AAA25049) from *Klebsiella pneumoniae* subsp. aerogenes, AlyDW11 (AEO50363) from uncultured bacterium, AlyB (AIY22661) from *Vibrio* sp. W13, and Aly7B (ANW96808) from *Wenyingzhuangia fucanilytica* CZ1127.

**Figure 4 ijms-25-05801-f004:**
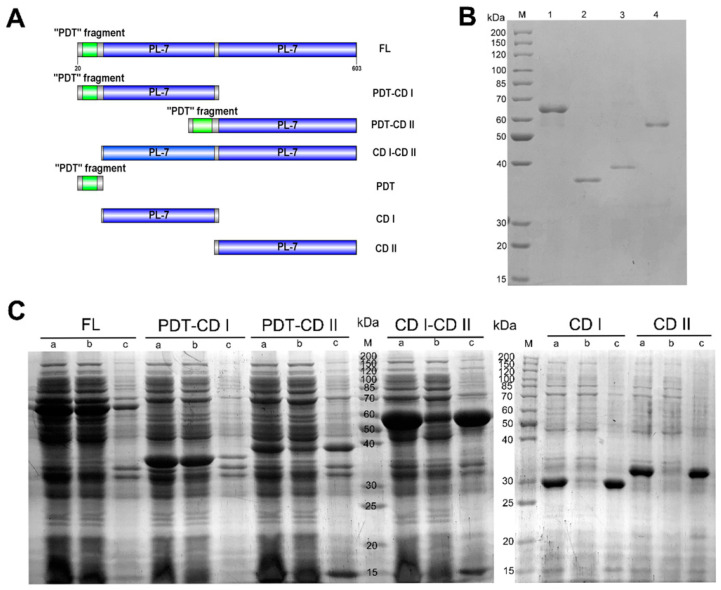
Design and recombinant expression of full-length VsAly7A and its truncated mutants. (**A**) Domain structure of full-length VsAly7A and its truncated mutants. (**B**) SDS-PAGE illustrating the full-length and other soluble truncated mutants. Lane M, molecular weight markers. Lane 1, VsAly7A-FL. Lane 2, PDT-CD I. Lane 3, PDT-CD II. Lane 4, CD I-CD II. (**C**) SDS-PAGE displaying FL VsAly7A and its truncated mutants under optimal induction conditions. Lane M, molecular weight markers. Lane a, induced cell lysate of *E. coli* cells. Lane b, supernatant fluid of the induced cell lysate. Lane c, precipitate of the induced cell lysate.

**Figure 5 ijms-25-05801-f005:**
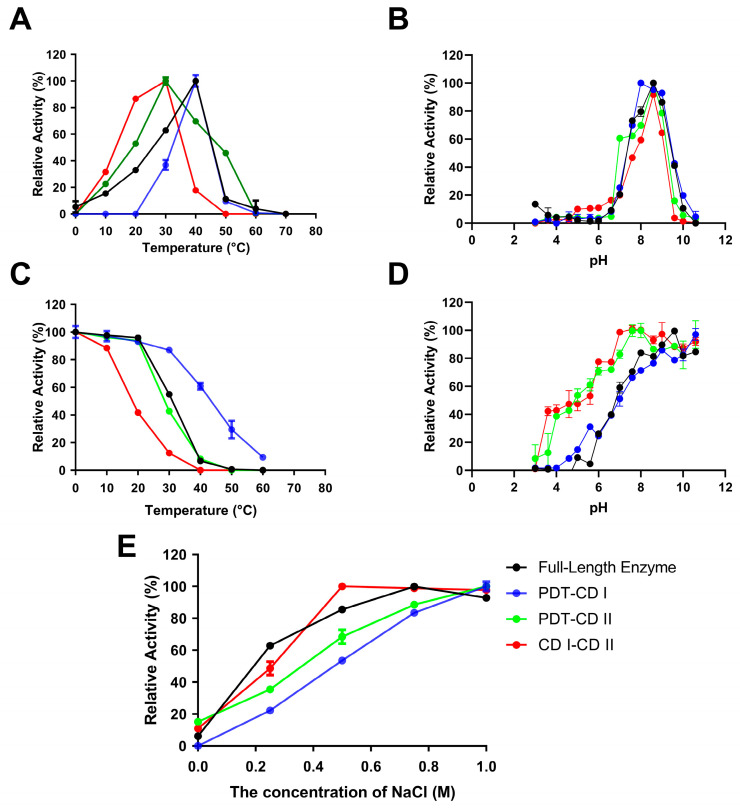
Biochemical characterization of VsAly7A and the four soluble truncated mutants. Optimal temperature (**A**), optimal pH (**B**), thermal stability (**C**), pH stability (**D**), effects of NaCl concentration (**E**) of FL VsAly7A and the four soluble truncated mutants. Values represent the mean of three replicates ± standard deviation.

**Figure 6 ijms-25-05801-f006:**
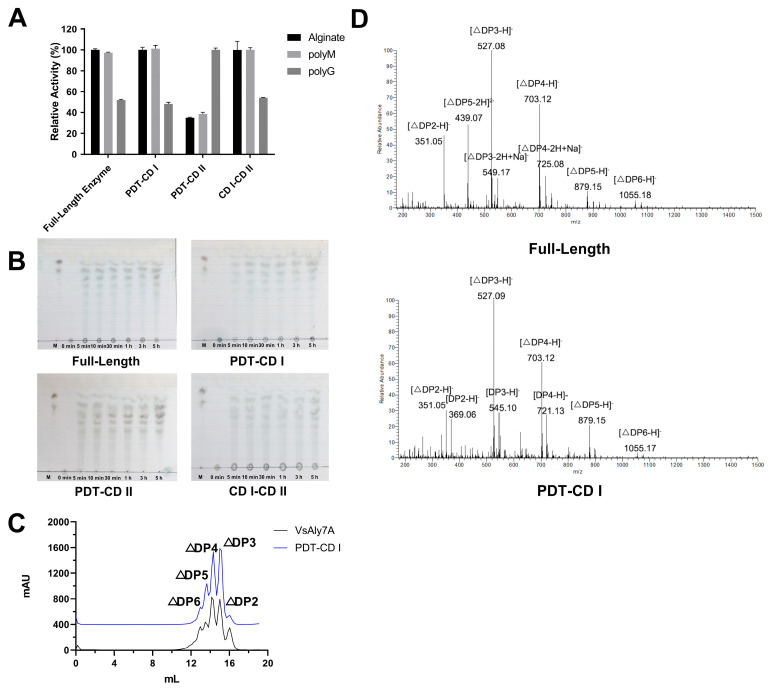
Substrate specificities and degradation of VsAly7A and its four soluble truncated mutants. (**A**) Substrate specificities of VsAly7A and its truncated mutants. (**B**) The time course of alginate degradation by FL VsAly7A and its truncated mutants were determined using TLC. (**C**,**D**) Analysis of end products of VsAly7A and PDT-CD I. (**C**) FPLC chromatogram. (**D**) ESI-MS analysis. The elution volumes of 16.04, 15.01, 14.16, 13,65, and 12.97 mL corresponded to unsaturated dis-, tri-, tetra-, penta- and hexasaccharide, respectively.

**Figure 7 ijms-25-05801-f007:**
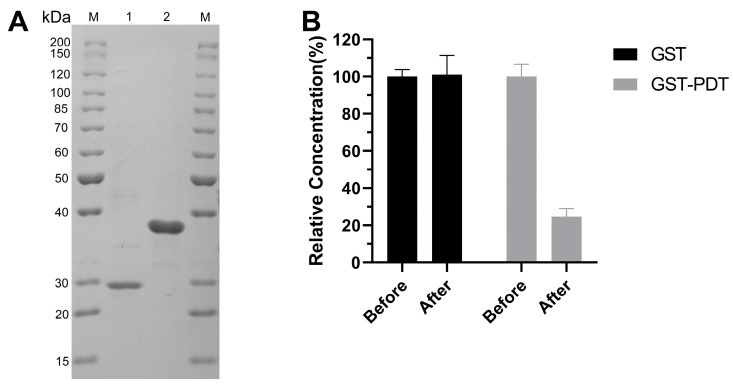
The alginate binding ability of GST and GST-PDT. (**A**) SDS-PAGE showing GST and GST-PDT. Lane M, molecular weight markers. Lane 1, GST. Lane 2, GST-PDT. (**B**) The different binding capacity of GST vs. GST-PDT with sodium alginate. Before, before centrifugation. After, after centrifugation.

**Table 1 ijms-25-05801-t001:** Specific activities of full-length VsAly7A and its truncated mutants.

Protein	Molecular Weight (kDa)	Specific Activity (U/mg)	Specific Activity (U/μmoL)	V_max_ (nmol·s^−1^)	*K_m_* (mM)	*k_cat_* (s^−1^)	*k_cat_*/*K_m_* (mM·s^−1^)
Full-Length	65.27	302.65	19,753.38	0.61	0.31	4.35	7.02
PDT-CD I	32.73	1955.00	63,992.15	0.49	0.29	13.30	23.33
PDT-CD II	39.11	70.13	2742.81	0.87	0.42	0.46	2.88
CD I-CD II	60.27	31.42	1893.62	0.76	4.05	0.03	0.006

**Table 2 ijms-25-05801-t002:** Primers used in this study.

Primers	Sequence (5′ to 3′)	Usage
Expression of VsAly7A and truncated proteins		
VsAly7A-FL-F	GGAGATATACATATGGGCGGCAGCAGCTCT	Expression of VsAly7A-FL
VsAly7A-FL-R	GTGGTGGTGCTCGAGTTGGTGACGGGTGCT	
PDT-CD I-F	GGAGATATACATATGGGCGGCAGCAGCTCT	Expression of PDT-CD I
PDT-CD I-R	GTGGTGGTGCTCGAGCCACGCGATTGAATC	
PDT-CD II-F	GGAGATATACATATGGGCGGCAGCAGCTCT	Expression of PDT-CD II
PDT-CD II-O-R	ATCTGTGTGTTCGATAATGTCTTGAAATTTCGTTATGGAGTACG	
PDT-CD II-O-F	TAACGAAATTTCAAGACATTATCGAACACACAGATTCAATC	
PDT-CD II-R	GTGGTGGTGCTCGAGTTGGTGACGGGTGCT	
CD I-CD II-F	GGAGATATACATATGCCGTACTCCATAACG	Expression of CD I-CD II
CD I-CD II-R	GTGGTGGTGCTCGAGTTGGTGACGGGTGCT	
PDT-F	GGAGATATACATATGGGCGGCAGCAGCTCT	Expression of PDT
PDT-R	GTGGTGGTGCTCGAGGCCTGAATTGTCTAA	
CD I-F	GGAGATATACATATGCCGTACTCCATAACG	Expression of CD I
CD I-R	GTGGTGGTGCTCGAGCCACGCGATTGAATC	
CD II-F	GGAGATATACATATGAAAATCGAACACACA	Expression of CD II
CD II-R	GTGGTGGTGCTCGAGTTGGTGACGGGTGCT	

## Data Availability

The sequence of VsAly7A was submitted to GenBank under Accession number OR921196.

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
