# Peer review of "A “Pro-Asp-Thr” Amino Acid Repeat from Vibrio sp. QY108 Alginate Lyase Exhibits Alginate-Binding Capacity and Enhanced Soluble Expression and Thermostability"

_ijms, 2024, doi:10.3390/ijms25115801_

Round 1

Reviewer 1 Report

Comments and Suggestions for Authors

The manuscript “A PDT amino acid repeat from Vibrio sp. QY108 alginate lyase exhibits alginate-binding capacity and enhanced soluble expression and thermostability” is interesting. The PDT fragment was found as a functional peptide. The research investigated the function of different domains of alginate lyase. It is meaningful. There are some issues in the manuscript:

1.     When expressing each type of protein, it is necessary to optimize the conditions. When comparing the expression of the four proteins, have the expression conditions been optimized separately, such as expression time and the amount of IPTG. So, did you optimize the condition?

2.     E. coli should be written as italic.

3.     Figure 3 is mistake with tag of ABC. Why the thermal stability was affected? Only by PDT fragment?

4.     Figure 4D was too small to see clearly.

5.     Abbreviations should be accompanied by an appendix in the manuscript.

6.     In figure 5, GST is 30 kDa? GST-PDT is 38 kDa?  

Comments on the Quality of English Language

Minor editing of English language required.

Author Response

Dr. Zheng Fu

School of Medicine and Pharmacy

Ocean University of China

5 Yushan Road, Qingdao 266003

P.R.China

Reviewer

Mar. 11th, 2024

Dear Reviewers:

    Sincere thanks should be given to the reviewer for the constructive comments and suggestions. The responses to the comments are given below, please see the attachment. Manuscript revisions have been marked in bold red font.

Reviewer 2 Report

Comments and Suggestions for Authors

In the present work authors reported and characterized a new algiante lyase (VsAly7A) enzyme from Vibrio sp. QY108, with two catalytic domains and a PDT fragment. The authors also identified that the PDT domain plays crucial role in soluble expression of the protein, binding affinity to substate and could be used as fusion tag to improve soluble expression of recombinant proteins. On the other hand, among the two domains CD-I exibited greater catalytic efficiency than CD-II domain, futher expressing it as a PDT-CD-I improve the thermal stability of VsAly7A. The findings are useful and demonstrated well. overall the manuscript is on boarderline to me but worth publishing of the readers of IJMS. before publishing it this reviewers has follwing questions to be addressed.

1) Line 49 and several other places in the manucript---> E. coli should be italic

2) Figure 1B, 1C and 1D is difficult to read and are of low resolution. could you please update these figures. 

3) if possible could you please use another synonym for the word "depicted" sounds unfamilier while reading.

4) In Figure 4D, it is difficult to read the values and fragmentation pattern? Please provide better resolution figures

5) In Figure 5B could you please modify the lables on x-axis, seems to be missleading?

6) Could you change the key word "cold-adapted" to somthng other like  ---------> "soluble expression"

7) Could please add a reaction scheme/figure showing catalytic reaction of Alginate lyases cleaving the 1,4-glycosidic bond..it might more informative?

8) Line 76-77---> It says that PDT has nine "Pro-Asp-Thr"  repeating fractments, however the parentheses contain "(Pro34-Tyr61)" could please double check and explain this part. I was not able to understand this explaination?

9) In general, could you please write short sentences in the main content------> longer sentences makes difficult to follow the content. Manuscript need to be improved for english

10) please rewrite this sentence?

11) please check for the references section----> in few references "journal issue" is provided and others does not have such inforamtion.  Please check for the journal abbriviations?  

Comments on the Quality of English Language

Could you please write short sentences in the main text. The longer sentences makes difficult to follow the content and raises ambiguity to readers. 

Author Response

(The authors gave the same response as above.)

Reviewer 3 Report

Comments and Suggestions for Authors

This is a good study on the contribution of different domains to substrate binding, activity, and stability by generating various deletion mutants. What the manuscript is lacking is the “science” behind the exciting results. There are so many things authors could have done to answer some questions that arose during this study. I have given certain comments to improve the manuscript. I recommend authors consult the publications on enzymes by Cavicchioli, R and Feller, G groups for further insight and interpretation of their data and any further experiments.

1.       Give EC number of the enzyme in Introduction Section.

2.       Figure 1 is unreadable.

3.       Authors MUST show the AlphaFold structures of the various forms of the enzyme used in this project. They can use ChimeraX to predict the structure or fetch it from the database if someone has already done it. In structure, show various domains in different colors.

4.       Single catalytic proteins unfold reversibly whereas multi-domain proteins unfold irreversibly. By removing CD2 domain, is the enzyme still unfolding reversibly or irreversibly? Is there any data to confirm this in this case? This is very easy to confirm experimentally by activity assays or using DSC, circular dichroism etc. The easiest would be activity and/or fluorescence by unfolding at a certain temperature and then cooling it and redoing the melting curve. By comparing Fluorescence/DSC results with the activity results, authors can infer whether the unfolding is Global or unfolding starts from the activity site (local)?

5.       Comparing Figures 3C and D reveals that the Topt of both FL and CD1 is 40 deg C (3C), but the Tm of FL is 10 deg C less than CD1. This shows that the in case of CD1, the unfolding of global structure is concurrent with the loss of active-site as its Topt (3C) and Tm (3D) are almost same. In contrast, the Topt of FL is 10 deg more than its Tm, suggesting that its active site is collapsing before the unfolding of overall structure. Discuss this in revised version.

6.       CD1 is very interesting as it shows that the removal of CD2 not only increases kcat but also melting temperature defying the activity-stability tradeoff. Generally, activity is increased due to increased FLEXIBILITY that concomitantly decreases the stability. Flexibility is very easy to determine using acrylamide solution.

7.       What is the function of CD2 domain in the function of the enzyme? Why it was inhibiting the enzyme? why its removal increased thermostability. The generation of structures by AlphaFold can answer the question.

8.       Line 2, Abstract: “render it a promising candidate for future applications,” In view of this statement, authors can do a PRODUCTIVITY experiment and compare FL vs CD1. Authors are referred to consult this recent paper.

9.       Abstract, line 12-13: Alginate lyases cleave the 1,4-glycosidic bond of alginate by eliminating sugar molecules from its bond. What is meant by “eliminating sugar molecules from its bond”?

Comments on the Quality of English Language

minor corrections

Author Response

(The authors gave the same response as above.)

Round 2

Reviewer 3 Report

Comments and Suggestions for Authors

The authors have revised the manuscript in view of the comments without carrying out extra experiments which I suggested. 

Comments on the Quality of English Language

Minor

Author Response

Dr. Zheng Fu

School of Medicine and Pharmacy

Ocean University of China

5 Yushan Road, Qingdao 266003

P.R.China

Reviewer

May. 10th, 2024

Dear Reviewers:

    Sincere thanks should be given to the reviewer for the constructive comments and suggestions. This response provides additional experimental data as mentioned by previous reviewers, please see the attachment.
